# Contribution of Human Pluripotent Stem Cell-Based Models to Drug Discovery for Neurological Disorders

**DOI:** 10.3390/cells10123290

**Published:** 2021-11-24

**Authors:** Alexandra Benchoua, Marie Lasbareilles, Johana Tournois

**Affiliations:** 1Neuroplasticity and Therapeutics, CECS, I-STEM, AFM, 91100 Corbeil-Essonnes, France; mlasbareilles@istem.fr; 2High Throughput Screening Platform, CECS, I-STEM, AFM, 91100 Corbeil-Essonnes, France; jtournois@istem.fr; 3UEVE UMR 861, I-STEM, AFM, 91100 Corbeil-Essonnes, France

**Keywords:** pluripotent stem cells, high-throughput screening, drug discovery, precision medicine, neurodegenerative diseases, psychiatric diseases, rare diseases, neurons, glia

## Abstract

One of the major obstacles to the identification of therapeutic interventions for central nervous system disorders has been the difficulty in studying the step-by-step progression of diseases in neuronal networks that are amenable to drug screening. Recent advances in the field of human pluripotent stem cell (PSC) biology offers the capability to create patient-specific human neurons with defined clinical profiles using reprogramming technology, which provides unprecedented opportunities for both the investigation of pathogenic mechanisms of brain disorders and the discovery of novel therapeutic strategies via drug screening. Many examples not only of the creation of human pluripotent stem cells as models of monogenic neurological disorders, but also of more challenging cases of complex multifactorial disorders now exist. Here, we review the state-of-the art brain cell types obtainable from PSCs and amenable to compound-screening formats. We then provide examples illustrating how these models contribute to the definition of new molecular or functional targets for drug discovery and to the design of novel pharmacological approaches for rare genetic disorders, as well as frequent neurodegenerative diseases and psychiatric disorders.

## 1. Introduction

Neurological disorders, such as neurodegenerative diseases and psychiatric disorders, are significant healthcare concerns since they trigger severe impairments in quality of life and have a large world prevalence [1]. The causes behind these diseases are multifactorial and not well understood. Neurological disorders are chronic with devastating consequences that can continue for years after diagnosis. Many insurmountable ethical and practical obstacles exist to conduct research on human subjects and primary brain samples, so experiments and conclusions have mainly relied on animal models both in vitro and in vivo. However, differences in physiology, genetics and developmental patterns between human and animal brains have led to discordance between preclinical drug studies and clinical trials [2,3,4,5,6,7]. Implementing relevant yet flexible human cellular models would help conducting drug discovery and hopefully decrease this high clinical failure rate.

Human pluripotent stem cells (PSCs), with their ability to self-renew and then differentiate into different types of brain cells, represent such an opportunity [7]. In 1998, Thomson et al. published, for the first time, a methodology for isolating and culturing human ESCs from blastocysts [8]. These embryonic stem cells (ESCs), which need to be harvested from human embryos, represented a valuable but very limited resource for human disease models. In 2007, Shinya Yamanaka’s team showed that PSCs could be reprogrammed from a small sample of skin fibroblasts by expressing, combined, the four pluripotency-associated transcription factors SOX2, OCT4, KLF4and c-Myc. To differentiate them from genuine PSCs derived from embryos, he named them induced pluripotent stem cells (iPSCs, [9]). Since then, more studies have described the successful reprogramming of iPSCs from different somatic cell types such as blood monocytes and epithelial urine cells [10,11,12,13]. Although there may be some methylation profile differences between iPSCs and ESCs due to the reprogramming process, they are considered equivalent regarding cell morphology, proliferation and differentiation capacity [14,15]. Importantly, iPSCs can be derived from patients who have neurological disorders, allowing researchers to study nervous system diseases within an endogenous human genetic background. The development of PSC-based technologies offers unique possibilities to investigate disease progression and perform drug discovery studies.

## 2. Integration of PSC-Derived Models in The Process of Drug Discovery

### 2.1. Strategies for Drug Discovery

Historically, therapeutic compounds were discovered by identifying the active ingredient in traditional remedies and testing those with given drug activity against pre-identified biological targets that were hypothesized to be disease modifiers, using the so-called candidate drug-based strategy. Progress in molecular biology and lab automation has progressively led to more systematic and agnostic approaches, including compound library screening, which consists of testing collections of thousands of small synthetic molecules or natural compounds with a cellular model. Cell-based drug screenings can be conducted at a single molecular target level, at a pathway level or at a phenotypic level [16,17]. Molecular target-based screening, was, until recently, the prevalent model used. In this approach, molecular targets are identified by basic research studies on disease models. The targets are generally gene products such as mRNAs or proteins that are abnormally expressed in a pathological context and are demonstrated to influence disease emergence or progression. Such molecular-based target screenings are usually developed in the context of monogenic diseases, in which the function of only one gene product is involved and possibly corrected, or if familial forms of more complex diseases exist and there is clear target identification [18,19,20]. Robust biochemical assays are then developed in order to identify hit compounds among large libraries using high-throughput screening techniques (HTS). An alternative approach consists of identifying pathways or molecular signatures rather than single-gene products that can be targeted by compounds [21]. The use of omic techniques, such as transcriptomics, proteomics and metabolomics, can be integrated to identified pathways that are associated with pathogenesis [22,23,24,25,26]. The epigenetic dimension can also be investigated in complex, non-Mendelian disorders by unbiased techniques to identify novel pathways that can serve as biological targets for HTS [27,28]. Interestingly, these pathways can be common to several diseases [29,30,31]. Finally, a phenotypic-based approach can be applied. Compounds are screened for their ability to normalize functional or phenotypic parameters in disease-relevant models, such as axonal transport, growth processes, synaptic functions or neurodegeneration. Compared with molecular and pathway approaches, functional screening represents a higher level of complexity; however, this integrated and agnostic strategy is also more promising for disorders of unknown, complex or multigenic origin. In phenotypic cell-based screenings, the assay is performed on a support that is suitable for image-based high-content screening (HCS) to identify positive hits [32,33,34].

Overall, the process of de novo drug discovery screening of new chemical entities (NCE), from the initial HTS or HCS to the final marketing of a compound, requires 10–17 years. To limit cost and decrease time, one possible strategy is to use specific chemical libraries that are smaller, but enriched with high drug-likeness compounds [35]. These compounds have already been tested for their safety and bioavailability and most correspond to drugs that are commercially available for human use. Positive hits represent candidates for drug-repositioning approaches or starting points for the discovery of new drugs, based on their known structure and mechanisms of action [36,37,38]. Compared to de novo drug discovery, drug repositioning allows the timeline for marketing a given compound to be shortened to 3–5 years. Drug repositioning is particularly attractive for rare genetic diseases [39,40] and for the identification of new molecules for subgroups of patients who do not respond to gold standard treatments or suffer many unwanted side effects that force treatment withdrawal.

### 2.2. Pluripotent Stem Cells as Biological Material

While extremely powerful, cell-based high-throughput screening (HTS) requires a large quantity of cells relevant to the disease that can be produced repeatedly, robustly and homogeneously. Ideally, these cells can be frozen to constitute large banks, in order to allow sequential runs of screenings to be performed and selected compounds to be further validated and characterized. Differentiated cells obtained from patient-derived PSCs offer both flexibility and relevancy for model disorders that affect brain development and functioning. Indeed, it is possible to obtain large quantities of cells without genetically modifying them with oncogenes thank to their self-renewal ability. Then, it is possible to differentiate PSCs into a variety of neuronal subtypes as well as glial cells. Protocols of differentiation are based on the recapitulation of human neural development in vitro (Figure 1). PSCs are exposed to developmental cues to guide them progressively along the steps necessary to specify a particular brain cell type. The first step consists of restricting the potency of the cells to the neural lineage to obtain neuro-epithelial structures resembling the embryonic neural tube. One of the most widely used and simple technique is the so-called dual SMAD inhibition, in which inhibitors of BMP and TGF-beta pathways are used to convert homogeneously PSCs into neuro-epithelial structures, named rosettes. The simple inhibition of these two pathways was demonstrated to be sufficient to release the cells from pluripotency while blocking their engagement in alternative fates [41,42]. Regional specification of the neuro-epithelial structures is obtained in a second effort by using patterning factors or morphogens. Dorso-ventral patterning is conditioned by the gradual and antagonistic role of BMPs and SHHs, while rostro-caudal positioning is under the control of the Wnt/beta catenin and FGF8 pathways [43,44]. The population of regionally restricted neural progenitors can be further amplified and frozen or terminally differentiated into post-mitotic neurons expressing brain region-specific markers and corresponding neurotransmitters identities [45].

This strategy was successfully applied to differentiate PSCs into a variety of neuronal subtypes relevant to neurodegenerative and neuropsychiatric diseases [46,47], such as hippocampus CA3 pyramidal neurons, which exhibit the electrophysiological properties of mossy fibers of the dendate gyrus [48]; hypothalamic-like neurons capable of secreting orexigenic and anorexigenic neuropeptides and responding appropriately to the metabolic hormones ghrelin and leptin [49]; GABAergic interneurons of the cortex and the basal ganglia [50,51]; serotoninergic neurons of the raphe nuclei [52,53]; dopaminergic neurons of the substantia nigra [44,54]; cortical pyramidal neurons [55,56,57]; and spinal motoneurons secreting acetylcholine [58,59]. Glial cells can also be derived from PSCs, including astrocytes [60,61,62,63,64], oligodendrocytes [65,66,67,68] and microglia [69,70,71,72], allowing the co-culture and recapitulation of cell-autonomous and non-cell-autonomous mechanisms leading to disease progression to be conducted [73,74,75,76,77,78,79,80,81,82]. Next to these “growth-factor”-mediated protocols, several groups have described methods of direct conversion of PSC into sub-types of neurons using forced expression of transcription factors. These protocols by-pass essential steps of neural differentiation and have the advantage to reduce the time of differentiation and homogenize neuronal production. Forced expression of NEUROG2 promotes the conversion of PSC into excitatory glutamatergic neurons [83]; ASCL1 and DLX2, with or without LHX6, into GABAergic neurons [84,85]; ASCL1, with NURR1 and LMX1A, into dopaminergic neurons [86]; and NEUROG2, with ISL1 and LHX3, into motoneurons [87]. Considerable efforts have also been made to standardize cultures of PSC-derived brain cells into miniaturized formats, such as 384-well plates, allowing a systematic analysis of pathological phenotypes and compound testing at large throughput to be performed [56,88,89,90,91,92,93,94,95,96,97,98].

In summary, compared to other models previously used for neurological disorder-related drug discovery, PSC-derived models combine the relevancy of human primary neural cells and the flexibility of a cell line. Progenitors can be expended until reaching the cellular mass necessary to perform target identification and to conduct high-throughput screening of compounds. Once differentiated, patient-derived cells possess the molecular, electrophysiological and morphological particularities of post-mitotic neurons of the patient, offering the possibility to study the influence of their genetic background on the expression of the disease and the differential response to drugs. In the last decade, PSC-neural cells derived from patients affected by rare genetic diseases have paved the way for disease modelling and drug discovery, mainly by examining the repositioning of existing therapeutic compounds. Collections of iPSCs reprogrammed from individuals with more frequent disorders are now available and offer an unpreceded substrate to better understand more complex diseases, stratify patients, develop new early pharmacological interventions and adopt precision medicine approaches (Figure 2).

## 3. Paving the Way: Rare Genetic Diseases

Rare diseases are conditions that affect a small proportion of the population (fewer than 200,000 persons in the USA or fewer than one in 2000 in Europe). The Orphanet portal for rare diseases and orphan drugs (http://www.orpha.net, accessed on 15 September 2021) currently lists more than 5800 rare diseases. Many are genetically inherited and the genetic causes are clearly identified [99]. From the beginning of the human PSC history, rare genetic disorders have been attractive models for proof-of-concept studies of disease modelling. hESC derived from embryos after pre-implantation genetic diagnosis were a first source of PSCs with natural, disease-inducing mutations. iPSCs now represent an opportunity to create collections of cells from cohorts of patients with clear genotype–phenotype correlations. Since only one gene product is involved, it is also a straight-forward strategy to correct or induce mutations by genetic engineering for in vitro validation, in a given cell type, of the genotype-phenotype correlation. Accordingly, the first “pharmacological” studies using PSC-derived neurons were aimed at demonstrating that several genetic diseases hallmarks were recapitulated successfully in hESCs or iPSC-derived neurons and that they could serve to validate candidate drugs at low throughput [100,101,102,103]. PSCs from patients with rare genetic diseases were then the first to be used as proof of concept that PSC-derived cells can serve as support to develop innovative compound identification strategies using HTS. In monogenic diseases, only one gene product acts as the relevant disease-modifier, implying a very clear path to the development of a screening assay with only one molecular target to focus on. Accordingly, most screening conducted relied on molecular target-based approaches and used libraries containing known bioactive compounds, in order to identify lead compounds or new therapeutic targets or approved drugs for direct repurposing. Several HTS experiments were successfully conducted using PSCs from patients with neurological disorders. All of these studies illustrate the invaluable advantage of working with cell types relevant to the disease.

### 3.1. Fragile-X Syndrome (FXS)

The first advantage is to be able to explore genes and pathways expressed at physiological levels without the need to introduce exogenous gene expression systems. This was exemplified in studies that screened compounds for genetic forms of ASD induced by the loss of function of FRMP [104,105,106] and duplication of a segment of chromosome 7 [107]. FXS was the first neurodevelopmental disorder to be modeled with this aim using iPSCs. FXS is a neurodevelopmental disorder characterized by mild-to-severe intellectual disability and abnormal behaviors, such as attention deficit, anxiety and depression [108]. FXS is also the most common known monogenic cause of autism, with 43–67% of male patients meeting the criteria of autism spectrum disorders. At the genetic level, FXS is linked to mutations (triplet repeats) in the 5′-untranslated region of the fragile X mental retardation 1 (FMR1) gene, which results in the absence of the FMRP protein. FMRP is a brain-specific RNA-binding protein that regulates the transport and translation of many mRNAs that play an important role in learning and memory [109]. Consequently, screenings were conducted in neural progenitors to identify compounds that could increase FRMP levels. Kaufmann and collaborators used FXS-patient iPSCs to develop an image-based HTS assay measuring the levels of FRMP in neural stem cells using immunofluorescence [104]. In all, 50,000 compounds were screened, including epigenetic regulators with known mode of action (7%), molecules covering a broad chemical space and biological diversity (46%) and a set of randomly selected compounds from an internal archive (47%). Four hits were identified and further confirmed for efficacy and absence of toxicity in dose-response experiments but were not further investigated for their mode of action or evaluated in an animal model. At the same time, Kumari and collaborators described the screening of 5000 known tool compounds and approved drugs in neural stem cells differentiated from an FXS patient-derived iPSC line using time-resolved fluorescence resonance energy transfer assay for FMRP detection [105]. Interestingly, the primary screening was performed in a 1536-well plate format, a format rarely used for cell-based assays and six compounds were identified that modestly increased FMR1 gene expression in FXS patient cells. Although none of these studies resulted in clinically relevant compounds, they provide strong proof of principle of the assays performed on patient-derived neural stem cells in a very high-throughput format to identify new lead compounds for FXS drug development. More recently, Li and collaborator used the newly described CRISPR/Cas9 system to create a reporter line for detecting FMR1 gene reactivation in human neural cells and used it to screen 1262 bioactive compounds [106]. This revealed two epigenetic regulators, 5-aza-dC and 5-aza-C, that significantly restored FRMP levels in disease cells. This study demonstrated that CRISPR/Cas9 can successfully be combined with iPSC-derived neural cells to design customized screening lines by knocking the luciferase reporter into endogenous target genes in order to obtain reporter lines and to reduce screening costs while increasing screening performance. This was possible only because iPSC-derived neural stem cells physiologically express endogenous levels of FRMP and proved the value of screening in human cells differentiated from PSCs. Together, these three studies demonstrate the feasibility and relevance of HTS in the neural progeny of PSCs for neurodevelopmental disorders.

### 3.2. Duplication of a Segment of Chromosome 7 (7Dup)

Validation that screenings can successfully be conducted for larger genetic aneuploidies was then reported. Duplication of a segment of chromosome 7 at 7q11 comprising 26–28 genes is one of the best-characterized copy number variations (CNVs) underlying autism. 7Dup patients show a range of autism spectrum disorder traits, especially varying degrees of language impairments and social restrictions [110]. Among the genes of the 7q11.23 region, general transcription factor II-I (GTF2I) has key relevance. This gene mediates signal-dependent transcription and plays a prominent role in various signaling pathways [111]. Most importantly, convergent evidences have implicated GTF2I as a major mediator of the cognitive–behavioral alterations in 7Dup [112]. Interestingly, deletion of this gene is also related to another rare disease, the Williams–Beuren syndrome. Cavallo and collaborators screened, using RT-PCR, 1478 compounds for their potential to increase GTF2I mRNA levels in 7Dup iPSC-derived cortical glutamatergic neurons. Some HDAC inhibitors were selected and further validated by quantifying the modulation of genes included in the segment duplication and involved in the Williams–Beuren syndrome [107].

### 3.3. Metabolic Disorders

Neurodegeneration is another aspect that requires authentic neurons to develop predictive models for drug screening, since these post-mitotic cells are more sensitive to metabolic stressors than peripheral cells. This was illustrated by studies of GM1 gangliosidosis and Lesch–Nyhan disease [113,114]. GM1 gangliosidosis is a lysosomal storage disorder characterized by abnormal accumulation of GM1 ganglioside. The main clinical feature of the disease is neural dysfunction due to massive GM1 ganglioside deposition in the central nervous system [115]. This abnormal deposition is caused by a deficiency in lysosomal β-galactosidase (β-GAL) activity which limits the body’s ability to degrade GM1 ganglioside in lysosomes leading to excessive GM1 ganglioside accumulation and eventual impairment of several pathways, including the unfolded protein response (UPR), endoplasmic reticulum calcium signaling and autophagy. Altogether, this induces progressive neurodegeneration. Kajihara and collaborators generated induced pluripotent stem cells (iPSCs) derived from patients with GM1 gangliosidosis, differentiated neurons and developed an image-based HTS assay to detect GM1 ganglioside accumulation. A collection of 2217 compounds containing already approved drugs and major chemicals used in pathway analyses was screened. The two best compounds, amodiaquine and thiethylperazine, were then shown to restore the presynaptic deficit in disease-derived neurons, upregulate the enzymes responsible for lysosomal glycosphingolipid degradation and activate autophagy. Interestingly, the authors also validated the hit compounds in a mouse model of GM1 gangliosidosis, demonstrating efficacy in reducing ganglioside accumulation in the brain and protecting it from degeneration [113].

LND is caused by deficiency of the purine salvage pathway enzyme hypoxanthine-guanine phosphoribosyltransferase (HGPRT), an X chromosome-encoded protein [116]. LND is characterized by severe neuropsychiatric disorders, which present with choreoathetosis, dystonia, aggression and self-injurious behavior [117]. Mutations in the HPRT1 gene, which code for HGPRT, are different for each individual but patients exhibiting the most severe neurological symptoms consistently have mutations that totally block protein synthesis [118]. To optimize the chance that a compound is efficient for most children with LND, independent of the type of mutation affecting HPRT1 gene, one strategy is to identify the compounds that activate alternative metabolic pathways that compensate for the deficiency of purine salvage in a target-agnostic manner. From a metabolic point of view, neural stem cells and neurons mainly rely upon recycling as a source of purine, while most other somatic cells rely, instead, upon de novo synthesis, a specificity that renders the brain more vulnerable to HGPRT deficiency than other organs [119]. In this context, the use of authentic human neural stem cells and neurons rather than peripheral cells such as fibroblast or blood cells was instrumental. Ruillier and collaborator decided to conduct a functional screening in neural stem cells and neurons derived from iPSCs of children affected with LND treated with azaserine, an inhibitor of the synthesis of purine de novo, in order to selectively induce cell death in HGPRT-deficient cells [114]. More than 3000 molecules were screened for their ability to rescue HGPRT-deficient cells from azaserine toxicity. Six pharmacological compounds were identified, all possessing an adenosine moiety, that corrected HGPRT deficiency-associated neuronal phenotypes by promoting metabolic compensations in an HGPRT-independent manner. Among these compounds, S-adenosylmethionine was reported in several case studies to ease the neuropsychiatric symptoms in LND [120,121,122,123,124], demonstrating the relevance of the screening strategy.

### 3.4. Cyclin-Dependent Kinase-Like 5 (CDKL5) Deficiency

Working with authentic and neuronal networks offers the opportunity for phenotypic and functional screening. This is of particular interest for diseases that involve abnormal excitability such as epilepsy. This was exemplified in a study by Negraes and collaborators, who conducted a phenotypic and target agnostic assay monitoring spontaneous calcium activity in 3D neuronal cultures as a read-out for network electric activity, which is abnormally increased in CDK5L-deficient neurons [125]. CDKL5 gene encodes for a serine/threonine kinase highly expressed in the central nervous system. Mutations in this gene cause CDKL5 deficiency disorder (CDD), characterized by neurodevelopmental delay, motor dysfunction, autistic features and early-onset intractable seizures, a defining trait that led to the standalone classification of this pathology [126]. Patients iPSC-derived 3D cortical spheroids exhibited hyperexcitability as measured as spontaneous calcium oscillations allowing a collection of 1112 compounds modulating different neuronal signaling pathways to be screened. Ivabradine, solifenacin, AZD1080 and crenigacestat were shown to reverse the phenotypic abnormality and were further investigated for their ability to ameliorate other CDD cellular phenotypes, including outward radial cellular migration defects. Due to their ability to regulate abnormal epileptic electrical activity in human neurons, these compounds open new therapeutic opportunities for other types of pathologies that include intractable seizures regardless of the initial trigger.

### 3.5. Phelan–McDermid Syndrome (PMS)

One final advantage of modelling neurological disorders with authentic neurons obtained from patient-derived iPSC is the development of personalized medicine strategies. This was achieved in PMS [127]. This is the first study in which a compound identified by HTS in a patient-derived cell line was actually evaluated directly and repurposed in the same patient. PMS is a neurodevelopmental disorder characterized by global developmental delay, intellectual disability, severe speech delays, poor motor tone and function, and ASD [128]. Genetic screening of the genome region identifies SHANK3 as the main gene involved in the ASD features associated with PMS. De novo truncating mutations inducing haploinsufficiency of the SHANK3 gene were estimated to be present in 0.69–2.12% of individuals with ASD [129,130]. SHANK3 is an abundant component of the postsynaptic density, where it acts as a scaffolding protein recruiting key post-synaptic elements, such as glutamate receptors, and linking them to the actin cytoskeleton [131]. Neurons differentiated from iPSCs of individuals with SHANK3 haploinsufficiency exhibited impaired electrophysiological responses to glutamatergic synapses’ stimulations, which could be corrected by re-introducing SHANK3 cDNA expression, validating their value for drug discovery [132]. In this study, iPSCs were derived from two children with PMS in order to constitute neuronal networks in a screening format (384-well plates). Patient-derived neurons exhibit reduced SHANK3 mRNA and protein expression, reduced neurite size, decreased glutamatergic synapses and decreased spontaneous network activity. In all, 202 marketed drugs were tested on these neurons and two of them, lithium and valproic acid, were demonstrated to increase SHANK3 levels (mRNA and synaptic protein), rescue neurite length and synapse numbers and, at least partially, restore network activity. Lithium was consecutively administrated during one year to one of the two patients and clinical examination showed significant improvement in the child’s social performance. This study demonstrated the feasibility of using patient derived-iPSC to select patient specific treatment, an approach described as personalized medicine.

Taken together, these studies using rare genetic disorders as models demonstrated that PSC-derived neural stem cell and neurons can be suitable biological materials to conduct compound screening at high throughput, can allow compounds that modulate endogenous targets that are not physiologically expressed in other cell models to be identified, are suitable for neurodevelopmental and neuropsychiatric diseases as well as neurodegenerative diseases and offer unique opportunities for phenotypic screening. This opens a path for research in prevalent multifactorial diseases and the promotion of precision medicine.

## 4. Neurodegenerative Diseases

Neurodegenerative diseases are among the leading causes of disability and a major cause of death worldwide. The first defining feature of these disorders is, of course, the death of neurons following a period of neuronal dysfunction and synaptic loss. The anatomical distribution of neurodegeneration determines the clinical pattern of individual disorders, which varies widely, but all of these disorders share progressive loss of cognitive and motor functions to varying degrees, which eventually leads to institutionalization and death. Pharmacotherapy is so far aiming at reducing clinical symptoms but does not stop disease progression. Misfolding, accumulation and aggregation of disease-specific proteins are universal features of neurodegenerative diseases that preceded the death of neurons by several years [133,134]. therefore, current investigations into therapeutic compounds are mainly focused on abnormal protein conformation and accumulation. However, PSC-based disease modelling additionally offers the possibility to follow disease progression and identify new molecular targets for earlier therapeutic interventions. They also offer the opportunity to develop strategies covering several disorders in a less specific manner. Examples of the contributions for the two most common neurodegenerative diseases, Alzheimer’s disease (AD) and Parkinson’s disease (PD), are provided.

### 4.1. Alzheimer’s Disease

AD is the most common cause of elderly dementia. This neurodegenerative disease is clinically characterized by progressive and gradual cognitive impairment, synapse loss and substantial loss of neurons in later stages. Currently, there are only two approved clinical treatments for AD, acetyl-cholinesterase inhibitors and N-methyl-D-aspartate receptor antagonists, both with very limited therapeutic effects. Early-onset AD is linked to autosomal-dominant inherited mutations in the genes encoding amyloid precursor protein (APP), presenilin 1 (PSEN1) and presenilin 2 (PSEN2). These cases are referred to as familial Alzheimer’s disease and are well characterized. In contrast, the etiology of 95% cases of late-onset AD, referred to as sporadic Alzheimer’s disease, is not known and may involve various triggers, including genetic and environmental factors [135]. Neuropathological hallmarks of AD are the formation of extracellular amyloid plaques, composed of aggregated amyloid β peptides (Aβs) and neurofibrillary tangles, formed by hyperphosphorylated tau proteins [136,137]. Both have received a great deal of attention in an effort to develop new pharmacological strategies [138]. Modelling of familial early-onset AD demonstrated that neurons differentiated from patient-derived iPSCs successfully recapitulated characteristic AD phenotypes, including the formation of Aβ aggregates and neurofibrillary tangles, which can be reversed by candidate pharmacological treatments, confirming the value of the model in the search for new pharmacological approaches [139,140,141,142,143]. Aβ accumulation and tau hyperphosphorylation were also reported using iPSCs derived from sporadic cases [144].

Aβ pathological peptides are produced by sequential cleavage of amyloid precursor protein (APP) by β-site APP cleaving enzyme 1 (BACE1) and γ-secretase. These two enzymes are currently the most investigated targets for disease-modifying drugs in AD. However, strong inhibition of γ-secretase and BACE1 widely perturbs the processing of numerous endogenous substrates important for physiological functions other than APP and causes serious side effects after long-term treatment. Phenotypic, target-agnostic screening of PSC-derived neurons could help to identify compounds with alternative modes of action, aimed at preventing Aβ accumulation and the resulting neuronal toxicity. Cell death induced by Aβ oligomers was one of the easiest read-outs to exploit for drug screening. Xu and collaborators screened a proprietary GSK library of several hundred compounds targeting kinase pathways using a luciferase-based method to quantify neuronal death induced in PSC-derived forebrain neurons by exposure to oligomers of β-amyloid 1-42 (Aβ1-42) proteins [145]. They identified 19 hits rescuing cell death and impaired neurite outgrowth deficits in this model. Mechanism of action studies demonstrated that the compounds targeted the CDK2 protein; therefore, this was proposed as a potent new target to protect neurons from Aβ toxicity in AD. The neuroprotective potential of CDK-2 targeting compounds was additionally confirmed in another HTS of 1000 compounds, spotting five hits acting as Cdk-2 modulators [146]. However, while successful, the screening cascades designed in these studies allowed the compounds blocking cell death downstream from Aβ aggregation to be identified, but they do not propose approaches aimed at identifying pathways that can be targeted to block early protein misfolding. Kondo and collaborators developed a high-throughput electrochemiluminescence assay to quantify the A42/A40 Aβ peptide ratio in neurons differentiated from iPSCs with PSEN1 G384A mutation [139]. They screened 1258 pharmaceutical compounds, acquired 27 primary Aβ-lowering hits, prioritized the hits by chemical structure-based clustering and selected six lead compounds. To maximize the anti-Aβ effect, they tested a synergistic combination of bromocriptine, cromolyn and topiramate as an anti-Aβ cocktail. The combination was evaluated in vitro in iPSC-derived neurons from 13 individuals, including familial and sporadic AD patients, confirming that the combination of anti-Aβ compounds could reduce Aβ aggregation in all participants. This study proposed the design of a new screening platform allowing researchers to systematically test compound libraries for efficacy against protein misfolding and aggregation in a mechanistic agnostic manner, which showed great potential for marketed drug repositioning, alone or in combination and to speed new chemical entity identification and development.

Lowering tau hyperphosphorylation is another attractive target for drug screening in AD. Several strategies have been proposed to evaluate a compound ability to reduce tau phosphorylation or, more globally, tau levels in neurons. Wang and collaborators [147] used iPSC-derived glutamatergic neurons combined with a high-content screening assay to identify tau-lowering compounds in the LOPAC library of bioactive compounds (>2000 compounds). They identified moxonidine and metaproterenol, two adrenergic receptor agonists, as a class of compounds with potential to reduce endogenous human tau levels and delay AD disease progression. Van der Kant and collaborators adopted a more specific approach by quantifying, using automated cell imaging, the level of pThr231Tau in neurons differentiated from iPSCs of patients with the familial form of AD consisting in the duplication of the APP gene [148]. A collection of 1684 approved and preclinical drugs was screened for their efficacy in lowering neuronal pThr231Tau using immunofluorescence to quantify the ratio of pThr231tau/total tau levels combined with cell viability. Four inhibitors of cholesterol synthesis, namely, atorvastatin, simvastatin, fluvastatin and rosuvastatin, showed the best activity on these read-outs. Mechanism of action studies pointed to 3-hydroxy-3-methylglutaryl-CoA reductase (HMGCR) inhibition and sequential proteasome activation as the main pathways involved in a compound efficacy. Consequently, the investigators selected another compound to reduce cholesterol esterase activity in cells, which was better tolerated by astrocytes and neurons. Efavirenz was tested using neurons derived from a cohort of patients with different forms of AD and demonstrated the same efficacy but better tolerability than the lead compound simvastatin. Altogether, this point to the cholesterol metabolism as a druggable axis regulating tau phosphorylation and, consequently, Aβ accumulation via activation of the proteasome. Together, these drug screening studies demonstrate new potential for marketed drugs in reducing Aβ toxicity in AD and point to CDK2 and HMGCR inhibition as new opportunities to develop more efficient treatments.

### 4.2. Parkinson’s Disease

PD is another well-described neurodegenerative disorder which affects over 6 million people worldwide, predominantly over the age of 65 [149]. PD is characterized by motor symptoms, including rigidity, resting tremor, bradykinesia and postural instability, and non-motor features, including cognitive impairments, anxiety and depression [150]. The motor symptoms are due to the progressive loss of dopaminergic neurons in the substantia nigra pars compacta, with approximately 50% of dopaminergic neurons lost in the midbrain at the onset of motor symptoms [151]. The majority of PD cases are idiopathic, with only about 10% attributed to heredity. Currently, pharmacologic treatments are aimed primarily at correcting dopamine insufficiency. However, an effective disease-modifying therapy has yet to be established. The pathological hallmarks of PD are the presence of intraneuronal cytoplasmic inclusions of α-synuclein (α-syn), named Lewy bodies, and dystrophic neurites that also contain α-syn deposits. The mechanisms leading to the formation and the pathogenic significance of these inclusions remain unknown. The reason why dopaminergic neurons are more prone to α-syn accumulation and vulnerable to its toxicity is not well understood, but PSCs offer a unique opportunity to model this phenomenon in a human genetic and epigenetic context. Several studies have reported the aggregation and toxicity of endogenous α-syn in neurons differentiated from PSCs of patients with familial forms of PD, which may be reversed by candidate drug approaches [152,153,154]. Increased levels of aggregated α-syn in PD suggest that defective protein handling and clearance contribute to its pathogenesis. Alpha-synuclein is degraded by both the ubiquitin-proteasome system and the autophagy/lysosomal pathway and they both represent attractive targets to modulate α-syn accumulation [155]. An analysis of dopaminergic neurons derived from patients with idiopathic and familial forms of PD identified that mitochondrial oxidative stress, leading to oxidized dopamine accumulation, resulted in reduced glucocerebrosidase enzymatic activity, lysosomal dysfunction and α-syn accumulation [156]. This toxic cascade was observed in human but not in mouse PD neurons and could be blocked by mitochondrial antioxidants and calcium modulators. This important link between mitochondrial and lysosomal/autophagy dysfunction in PD pathogenesis, revealed thanks to patient-derived PSCs, was the focus of the development of methods aimed at screening compound libraries for early intervention in PD [157]. In order to identify therapeutic agents to ameliorate mitochondrial clearance, Yamaguchi and collaborators used dopaminergic neurons obtained from patients with familial PD resulting from Parkin or PINK1 mutations [20]. The proposed system recapitulates the deficiency of mitochondrial clearance, ROS accumulation and increased apoptosis by treating of these neurons with carbonyl cyanide 3-chlorophenylhydrazone. Accordingly, the image-based high-content screening read-outs included the quantification of mitochondria size and area, reactive oxygen species production with ROS dyes and neuronal death monitored with caspase-3. In all, 320 pharmacologically active compounds were screened for their ability to ameliorate these parameters. Four hits, MRS1220, tranylcypromine, flunarizine and bromocriptine, were identified and further tested in idiopathic iPSC-derived neurons and *PINK1*-inactivated *Drosophila*. Bromocriptine was the most efficient compound. Interestingly, it was the second time that bromocriptine was identified by HTS on iPSC-derived neurons as a molecule lowering protein misfolding and aggregation, since it was one of the molecules proposed by Kondo as an anti- Aβ aggregation agent [158]. This demonstrates that neurons derived from PSCs can help in identifying common targetable pathways in PD and AD.

### 4.3. Screenings for Compounds Targeting Several Diseases

Strategies for screening based on protein misfolding as a read-out have identified independently common compounds that show promise for different disorders, suggesting that less specific, phenotypic-based approaches may also lead to drugs with a larger spectrum of action. Neurotrophic factor administration has long been proposed as a therapeutic option for diseases in which the main component is cell death [159]. Brain-derived Neurotrophic factor (BDNF) was one of the most investigated trophic factors [160,161]. BDNF is synthetized and secreted by neurons upon release of transcription from the repressing factor REST. With REST being expressed endogenously by cortical progenitors and neurons differentiated from PSCs, this model was used to screen new chemical entities for their ability to promote REST inhibition and endogenous BDNF synthesis [162]. HTS of a library of 6984 new chemical structures using a luciferase assay measuring REST activity in neural derivatives of hESC led to the identification of two benzoimidazole-5-carboxamide derivatives. The most potent compound, X5050, was found to target REST degradation, but not REST expression, RNA splicing or binding to the RE1 sequence. Differential transcriptomic analysis revealed the upregulation of neuronal genes targeted by REST in wild-type neural cells treated with X5050. This activity was confirmed in neural cells produced from iPSCs derived from a patient with Huntington’s disease. Acute intraventricular delivery of X5050 increased the expressions of BDNF and several other REST-regulated genes in the prefrontal cortex of mice with quinolinate-induced brain lesions. Altogether, this points to X5050 as a lead compound to be chemically optimized and evaluated in different models of neurodegenerative diseases. Astrocytes are another attractive target for drug discovery in neurological disorders. Astrocytes are the predominant cell type in the nervous system and play a significant role in maintaining neuronal health and homeostasis. Recently, astrocyte dysfunction has been implicated in the pathogenesis of many neurodegenerative diseases, including Alzheimer’s disease, Parkinson’s disease, Huntington’s disease and amyotrophic lateral sclerosis [163]. Reactive oxygen species (ROS) contribute to the progression of neurodegenerative disease, so preventing ROS-related astrocyte dysfunction and death could, in turn, help prevent neuron damage and death [164]. Similar to neurons, primary astrocytes are difficult to harvest for the adult brain and PSCs offer an attractive alternative. High-throughput phenotypic screening using human ESC-derived astrocytes was conducted with the aim of identifying compounds that could protect against oxidative stress [165]. Astrocytes were exposed to hydrogen peroxide and apoptotic nuclei quantified in 1536-well plates. In all, 4100 bioactive and approved drugs were screened and nine hits, including norcantharidin, tyrphostin A1, oxyphenbutazone and enzastaurin, were proposed as promising lead compounds for further optimization to protect from ROS-induced neurodegeneration. Finally, neuroinflammation, another hallmark of neurological disorders, is accompanied by the production of neurotoxic agents such as nitric oxide [166]. An HTS assay using a stem cell-based model of neurodegeneration induced by neuroinflammation was used to screen 44,000 new chemical entities form the LDC compound collection (Lead Discovery Center, Dortmund, Germany) and a family of small molecules was identified that shared the property of dually inhibiting both CDK-5 and GSK-3 [167,168]. These molecules protected the cytoskeleton of human neurons co-cultured with activated microglial cells and promoted survival. One compound, LDC8, showed promising results in a zebra fish model of AD. Recent models of co-culture of neurons and microglia derived from PSC should also provide useful biological substrates to screen for molecules modulating neuroinflammation-induced neurodegeneration [169].

Together, these studies demonstrate the contribution of PSC-derived models to the discovery of new axes of research for drug discovery for the two most common neurodegenerative diseases and illustrate how dynamic this new field is. However, neurons and glial cells differentiated from PSCs remain relatively immature in vitro due to the protracted time necessary to reach full functionality. This may be a limitation of these models to study age-related phenotypes, that, by definition, appear late in life and compromise translation into clinics. Efforts have been put to reduce the time of maturation of these cells by using protocols of direct conversion with forced expression of transcription factors, as described earlier in this review, but also by promoting emergence of age-related phenotypes by exposing cells to specific proteins, such as progerin, in order to fast-forward stem-cell aging [170].

## 5. Psychiatric Disorders

Psychiatric disorders, including major depressive disorders (MDD), schizophrenia (SCZ) and bipolar disorder (BPD), are estimated to affect one of every three individuals during their lifetime [171,172]. MDD is one of the most common psychiatric diseases. It is characterized by profound dysregulation of affect and mood and is also associated with other abnormalities, including cognitive dysfunction, sleep and appetite disturbance, fatigue and many other metabolic, endocrine or inflammatory alterations [173]. It has been reported that nearly 5% of the population in the developed countries meets criteria for MDD. MDD is a complex and multifactorial condition involving abnormal serotonin levels in the brain, which is the main target for current treatments, but not solely [174]. Other dysregulations in MDD brains include perturbation of neurotrophins synthesis and abnormal neurogenesis throughout the lifespan in the limbic system of affected individuals [175,176]. SCZ is a chronic and debilitating psychiatric disorder characterized by hallucinations and paranoid delusions (positive symptoms), apathy and anhedonia (negative symptoms) and disordered thought processes and working memory (cognitive deficits) [177]. SCZ affects approximately 1% of the general population. The first psychotic break happens in adolescence or early adulthood and it usually develops into a chronic condition requiring lifelong treatment [178]. SCZ is believed to be a disorder of abnormal neurodevelopment [179], but the cellular processes that lead to the onset and persistence of symptoms are unknown. Genome-wide association studies have identified a number of common variants that are associated with SCZ [180,181]. Some of the pathways reported as dysregulated in these studies are involved in neural differentiation, synaptic transmission and circuit development, giving weight to the hypothesis that SCZ is a developmental disorder. BPD, also known as manic–depressive illness, is a mood disorder that also affects approximately 1% of the population and is characterized by episodes of mania and depression [182,183]. While the heritability of BPD is very high, the causative biological factors are unknown and genetic studies have only recently led to some clues. BPD is typically diagnosed in adolescence or early adulthood, suggesting that, as for SCZ, abnormal neurodevelopment may play a role in its etiology and progression [184]. Common to all psychiatric disorders, scientific investigations have been hindered by a lack of relevant cellular and animal models. Part of the reason for this is a fundamental lack of knowledge of the genetic and molecular processes underpinning the appearance and persistence of symptoms that categorize these disorders. Another reason is the existence of major differences between humans and the animal species classically used as models in the pathways controlling the development of the neuronal circuits involved in higher brain functions, that are impacted. Patient-derived stem cells provide promising new ways to overcome the obstacles to studying live neurons in the lab, document disease-related signaling pathways and reveal valuable new strategies for drug repositioning or discovery [185].

### 5.1. Strategies Based upon Modulation of Neurogenesis

Most psychiatric disorders are classified as neurodevelopmental disorders that affect, more specifically, neuronal circuit formation and activity; in the absence of clear molecular mechanisms to target, the first drug discovery studies involving PSCs focused on phenotypic markers in a non-disease-specific context. Neurogenesis was one of the first physiological processes targeted by such phenotypic screenings. Hippocampal neurogenesis has been demonstrated to be central in MDD pathology in animal models and current antidepressant molecules have an activity on neurogenesis, including the gold standard fluoxetine [186,187]. Post mortem brains from BPD patients show enlarged ventricles, suggesting a loss of cortical volume, more particularly of grey matter, and volume abnormalities in the hippocampus. These volume abnormalities are ameliorated by lithium treatment [188,189,190,191,192]. A small but significant reduction in hippocampal volume was also reported in SCZ patients compared with healthy controls, primarily in the dentate gyrus (DG) and Cornu Ammonis 3 (CA3) hippocampal fields [193,194]. The process of neurogenesis includes neural stem cell proliferation, differentiation as neurons, then neurite outgrowth and synaptogenesis. These steps have been recapitulated in vitro using human PSCs in a format amenable to HTS. High throughput, image-based high-content analysis has been used successfully to monitor the rate of neural stem cell proliferation and their ability to differentiate as post-mitotic neurons, then develop as neuronal circuits. In at least two neuronal systems relevant to psychiatric disorders, glutamatergic neurons of the superficial layers of the cortex and cortical GABAergic interneurons, HTS has identified compounds of interest [56,92]. These studies screened collections of thousands of molecules consisting of approved drugs, well-characterized tool compounds, natural products and human metabolites. Among the hits identified, several tool compounds and approved drugs already in use for psychiatric diseases were shown to modulate neurogenesis under different aspects, including typical and atypical anti-psychotics and antidepressants. This suggests that the efficiency of these molecules may involve the modulation of neurogenesis in parallel or in synergy with their known role in dopaminergic and serotoninergic neurotransmission. In addition, screening of this type of collection provides evidence that many pathways that play role in neurogenesis, with regard to psychiatric disorder treatment, can be identified in iPSC-derived neurons using unbiased phenotypic screens, opening a path to discovering totally new chemical entities or repurposing and combining use of existing drugs.

### 5.2. Precision Medicine

The nosography of psychiatric disorders is mainly based on the appearance of core symptoms. The causes of symptom emergence and evolution are multiple and largely unknown. Patients vary widely in their clinical presentation along a spectrum, with some overlap between disorders and many comorbidities. In addition, common core symptoms can be attributed to different causes or result from different molecular dysregulations in the brain. Consequently, even with the same diagnosis, patients can vary in their patterns of therapeutic response to disease-specific treatments. The absence of a specific molecular signature implies delayed diagnosis, with different medications being tried on a patient until an effective regimen is identified empirically, which can often lead to disengagement and aggravation of the disease. Patient-derived neuronal circuits reconstituted in vitro from PSCs can help in identifying pertinent molecular signatures to help understand the sequence of events leading to the appearance of the symptoms and stratify the patients. The first step in such a study is to identify valid and robust disease-associated phenotypes so as to discriminate relevant patient subgroups, allowing the researcher to conduct a deeper investigation into the molecular characterization of these subgroups. Selective serotonin reuptake inhibitors (SSRIs), the leading antidepressant molecules, act by regulating serotonergic neurotransmission. However, SSRI resistance is observed in approximately 30% of MDD patients. Patient stratification based on pharmacological responsiveness combined with patient-derived neurons has been used to identify alternative and complementary pharmacological strategies [195,196]. A comparison of neurons derived from MDD-affected individuals, clinically classified as good responders or non-responders to classical SSRIs, identified molecular pathways discriminating non-responders that could be the basis for a more personalized approach. An analysis of serotoninergic neuron morphology demonstrated that neurons from non-responders showed abnormal neuritic networks, while serotonin synthesis and release were not modified. Using large-scale omic strategies, the authors linked the abnormal networks to decreased expression of two cytoskeleton-associated proteins, protocadherin A6 and A8 [195]. This indicated that molecules that restore protocadherin expression or, more indirectly, correct serotoninergic neuron morphology could be a potent alternative strategy in SSRI-resistant MDD patients. In parallel, a phenotypic analysis of the activity of forebrain glutamatergic neurons, one of the neuronal populations regulated by serotonin levels, highlighted that SSRI non-responsive neurons were hyperactive in the presence of serotonin, a phenomenon linked to the overexpression of two serotonin receptors, 5HT2A and 5HT7. Hyperactivity can be normalized using lurasidone, an FDA-approved selective 5HT2A receptor antagonist, indicating that this complementary pharmacotherapy can be adapted for some MDD patients [196]. Similar strategies were employed to model lithium resistance in BPD. Depending on the cohort studied, 30–50% of BPD patients did not respond to lithium, considered the gold standard of mood stabilizers [197,198]. Lithium resistance was modelled in hippocampal neurons that were differentiated from individuals with BPD and controls selected from two cohorts in order to examine neuronal action potential firing. This revealed a hyperexcitability profile in action potentials in immature neurons in BPD patients. Importantly, this in vitro hyperexcitable phenotype was reversed with lithium in neurons derived from patients who had responded to lithium treatment clinically, but not in neurons from patients who had failed to respond therapeutically to lithium. This suggests that hyperexcitability in immature hippocampal neurons may be a pathophysiological feature in BPD and could serve as a criterion to identify new therapeutic compounds [199]. Further analyses demonstrated that lithium treatment failed to activate LEF-1, a downstream effector of Wnt/beta-catenin signaling, in neurons from BPD patients who did not respond to it. Interestingly, valproic acid, used as an alternative to lithium, activated LEF-1 and was efficient in normalizing hyperexcitability in neurons of lithium-resistant patients [200]. This suggests that compound screening based on the LEF-1 reporter system could be a valuable tool to identify new treatments for BPD [201].

## 6. Concluding Remarks

PSC-derived neural cells offer a new and potent strategy to discover new pharmacological therapies for neurological diseases. Improved and standardized in vitro differentiation protocols, allowing a variety of otherwise inaccessible brain cell types to be studied, provide a foundation for establishing cellular models highly relevant to human pathologies. Proof-of-concept of the high transferability of PSC models in rare genetic disorders have been established. These studies have successfully demonstrated that this type of humanized cell-based model, anchored in a human disease-relevant cell type, could respond accurately to an endogenous stimulus such as a disease-causing mutation and a measurable, translatable endpoint that could also be monitored in patients. Successful repositioning of marketed drugs has strengthened these results. These experimental paradigms have been successfully applied to molecular target-based screening for the most prevalent devastating neurodegenerative diseases using patient-derived iPSCs obtained not only from familial cases but also, most importantly, from idiopathic cases. Next to the conventional molecular target-based approaches, patient-derived neuronal circuits have shown potential as biological tools for developing personalized or precision medicine approaches by opening the field of phenotypic screening in diseases previously considered inaccessible, such as psychiatric disorders, even in absence of clear molecular mechanisms involved in the emergence of the phenotype. PSC models are dynamic and allow researchers to reconstitute the progressive evolution of a disease, i.e., to dissect the sequence of molecular events leading from the initial trigger to the ultimate phenotypes (cell death in the case of neurodegenerative diseases, abnormal neuronal network activity in the case of neurodevelopmental disorders and psychiatric diseases). Each step can serve as a read-out for HTS. The typical screening cascade would be to perform the HTS on the earliest events, then to validate that the selected hits efficiently block disease progression by evaluating their efficacy on latter phenotypes.

In the future, the combination of PSCs with other powerful technologies will continue to show potential for drug discovery. Organoids and, more generally, 3D cultures, have begun to facilitate the generation of more mature and functional cell types (Sharma et al., 2020). Organoids generated with iPSCs recapitulate embryonic development to create self-organized 3D structures that replicate the complexity of specific organs. This technology can be beneficial for studying disorders in which developmental processes or multiple cell types within an organ are affected and for evaluating the potential of small molecules [98,202]. The CRISPR/Cas9 gene editing system has enabled unbiased and large-scale genetic perturbation screens to identify causative pathways by knocking out many genes in parallel and selecting cells with the desired phenotype. The combination of CRISPR screens with PSC technology would create a powerful tool to identify disease-causative or -modifying genes and pathways to be targeted by candidate drugs or involved in the mechanism of action of these drugs, at large scale and in an unbiased manner [203,204]. With the production of PSC-derived progenies being highly scalable, these cells can be used for comparative omic analysis, including bulk and single-cell RNA sequencing. This would enable detailed comparisons of controls and disease-affected cells to identify new pathways, to explain drug mechanism of action and to establish genetic profiles of good and bad responders to a given treatment. Finally, artificial intelligence and machine learning tools offer the promise of revolutionizing drug development [205]. With multiparametric profiling, the measured phenotypic changes in neuronal circuits can effectively serve as information packages summarizing cellular responses to pathological conditions or compound treatments. Artificial intelligence tools can harness the strengths of phenotypic assays leveraging the rich information reflected in phenotypic cellular changes to shed light on novel compounds and to expand our biological understanding. The expectation is that the implementation of these technologies over the past five years should begin to translate into an increased probability of success for individual drug discovery programs that use them. PSC models in drug discovery for neurological disorders is steadily transitioning from novel to mainstream.

## Figures and Tables

**Figure 1 cells-10-03290-f001:**
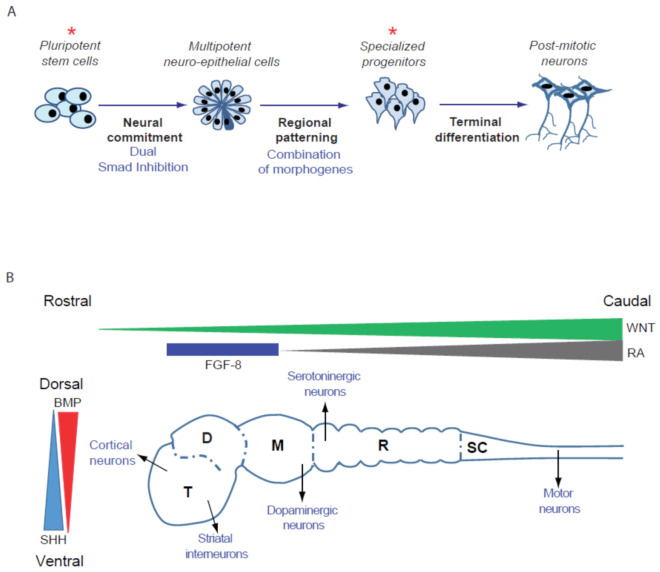
Specification of different types of neurons starting from pluripotent stem cells. (**A**) Steps leading from PSCs to terminally differentiated neurons. Stars indicate the self-renewing cell types that can be amplified and frozen to create large cryopreserved banks of biological materials. (**B**) Schematic representation of the human fetal brain with the 5 different regions, example of the type of neurons that emerge from these regions and the main morphogens involved in patterning. T, telencephalon; D, diencephalon; M, midbrain; R, rhombencephalon; SC, spinal cord; BMP, Bone Morphogenetic Factor; FGF-8, Fibroblast Growth Factor-8; RA, retinoic acid; SHH, Sonic Hedgehog.

**Figure 2 cells-10-03290-f002:**
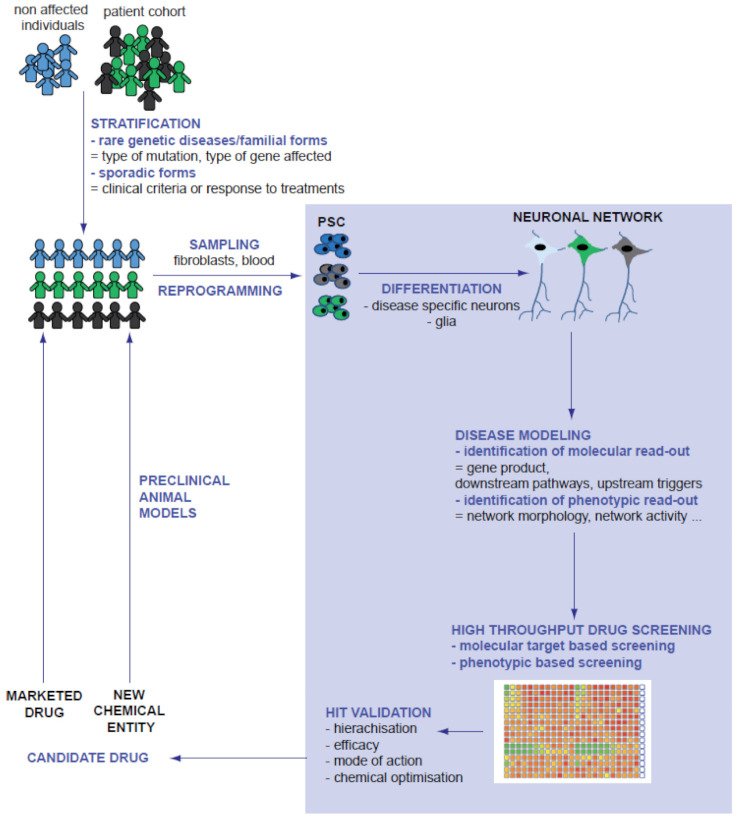
Integration of PSC-derived models in the process of drug discovery. Grey font highlights the part of disease modelling, drug discovery and compound optimization performed on PSC-derived cells.

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
