# Peer review of "Contribution of Human Pluripotent Stem Cell-Based Models to Drug Discovery for Neurological Disorders"

_cells, 2021, doi:10.3390/cells10123290_

Round 1

Reviewer 1 Report

The review gives a broad and comprehensive overview about the topic of iPSC-based drug research in neurological disorders. Several neurodegenerative/neuropsychiatric disorders and corresponding iPSC drug studies are described in detail, supported by an extensive list of references, allowing the reader to gain in-depth understanding about gold standards as well as recent discoveries in the field. The two figures offer a well-presented graphical summary of the implementation of iPSC in drug research and the generation of disease-relevant brain cell types from iPSC. In general, the review provides a clear overview of the very relevant field of in vitro pre-clinical drug testing using iPSC-derived cell types for both researchers in the field and a wider audience.

major points:

  1. For improved readability, main body of text should be shortened, especially neurodegenerative diseases/neuropsychiatric diseases. The paragraph describing iPSC-derived glial cells in the neurodegenerative section can be removed since glial cells are not mentioned in the neuropsychiatric part. The authors may consider further subheadings.
  2. In the abstract and in some text chapters “functional neuronal networks” derived from iPSC are mentioned. However, no publications were cited which show application of drugs to such fully functional circuits. This should be revised for a more cautious interpretation.
  3. only growth factor-based differentiation protocols of iPSCs-derived neurons were mentioned. Such protocols often suffer from low differentiation efficiency resulting in heterogeneous neuronal populations. It would be nice to mention direct differentiation protocols for example via the overexpression of lineage specific transcription factors.
  4. There is very little description of disadvantages/limitations of iPSC-based models. iPSC-derived neurons and glia may not be mature enough to model late-onset neurodegenerative diseases such as Parkinson’s and Alzheimer’s disease. How does this affect translation into clinics?
  5. Style, grammar, typos: There are several orthographic and grammatical issues throughout the manuscript. A major revision is mandatory. Here only few examples out of many:
  • Missing words and letters (e.g. line 33, line 168, line 187)
  • Incorrect usage of words (e.g. line 50 “differentiate”, line 308 “compassionate”)

Please check abbreviations are only introduced when words appear for the first time, not later. Use italic letters for “in vitro”. Pay attention to the correct usage of international notations of genes and gene products.

Reviewer 2 Report

Overall, this is a very thorough review describing the application of pluripotent stem cell models for drug discovery in neurodegenerative disorders and their potential to be used in personalized medicine, especially for complex neuropsychiatric disorders. Only minor changes are requested.

In the abstract the authors note the difficulty of modeling step-by-step disease progression for drug discovery, but in the cited cases, the HTS application to IPSCs focus on discrete phenotypes. Could the authors discuss the promise of these models to examine multiple stages of disease progression?

The authors should not the limitation of PSC models in capturing neurodegenerative phenotypes.

Could high throughput screening also be used to model blood brain permeability of specific drugs?

While the authors focus on neurons and astrocytes, given the high rate of inflammation in neurologic disorders, the authors should discuss the potential benefits of HTS screens in IPSC-derived microglia (as noted in PMC5482419) or at least note this omission in the discussion.

Line 53 – are the authors referring to epithelial cells in urine? 

Line 70 – missing commas

Line 119 – missing a space

Line 124 – thank should be thanks

Line 190 – In some genetic disorders, such as 22q11.2 and 16p11.2, multiple genes are impacted.

Line 212 – Remove ) after FXS

Line 221 – FSX should be FXS

Line 229 – What is a ‘tool’ compound?

Line 257 – Why is citation 103 in red?

Line 258 is missing a citation.

Cas9, Cas-9, SHANK3, SHANK-3 – be consistent with use of hyphens

Line 348 – to should be two

Line 456 – point should be points

Line 491 – sentence grammar needs to be corrected

Line 560 – sentence grammar needs to be corrected

Line 565 – ‘in the developed…” should be “in developed…”. Please remove the

Line 567 – foe should be for

Line 567 – sentence beginning with “Hypothesis include…” needs to be reworded

Line 592 – Remove extra period

Round 2

Reviewer 1 Report

The manuscript has been sufficiently revised